# Identity Management Systems for the Internet of Things: A Survey Towards Blockchain Solutions

**DOI:** 10.3390/s18124215

**Published:** 2018-12-01

**Authors:** Xiaoyang Zhu, Youakim Badr

**Affiliations:** 1INSA Lyon, University Lyon, LIRIS, UMR5205 Villeurbanne, France; 2Big Data Lab, Engineering Division, The Pennsylvania State University, Malvern, PA 19355, USA

**Keywords:** identity, decentralized, blockchain, access control, security, privacy, trust

## Abstract

The Internet of Things aims at connecting everything, ranging from individuals, organizations, and companies to things in the physical and virtual world. The digital identity has always been considered as the keystone for all online services and the foundation for building security mechanisms such as authentication and authorization. However, the current literature still lacks a comprehensive study on the digital identity management for the Internet of Things (IoT). In this paper, we firstly identify the requirements of building identity management systems for IoT, which comprises scalability, interoperability, mobility, security and privacy. Then, we trace the identity problem back to the origin in philosophy, analyze the Internet digital identity management solutions in the context of IoT and investigate recent surging blockchain sovereign identity solutions. Finally, we point out the promising future research trends in building IoT identity management systems and elaborate challenges of building a complete identity management system for the IoT, including access control, privacy preserving, trust and performance respectively.

## 1. Introduction

The Internet of Things aims at connecting everything ranging from individuals (human-beings), collectives (homes, organizations, companies, etc.) to things such as objects from physical and cyber worlds [1]. By 2020, there will be over 13.5 billion connected devices [2], which are equipped with sensing and actuating capabilities. Obviously, considering billions of people, trillions of IoT devices, and innumerable data resources, the major challenge is how to uniquely identify these entities and how to allocate digital identities to individuals and things through interconnected networks so that IoT entities can be easily identified and communicate with each other. Generally speaking, the goal of having an identity is for facilitating connection or communication among entities (e.g., human). A lone wolf does not need to rely on the identity too much in that living independently instead of with a group does not need to interact or communicate with others. Similarly, coupled with interconnections and interactions, digital identities of all the entities in IoT systems are the stepping stone to the digital world. Without digital identities, these IoT entities could barely transact with others, leading to untrusted environments and consequently the lack of business opportunities. This survey on the identity management, as an extension of our previous conference paper [3], covers many aspects from the identity originality of logic in the philosophy field, the identity solution analysis on the Internet and IoT, the new identity paradigm under the blockchain circumstance and the existing challenges of identity management in IoT.

Digital identities remain the keystone of online services and upon which security mechanisms (i.e., authentication, authorization, secure exchanges, etc.) and protocols are built in the Internet era. As defined by the International Telecommunication Union (ITU), an identity refers to a set of information used for uniquely identifying an entity in a given context [4], whereas an Identity Management System (IdMS) refers to the management of identity information through a set of operations, including registering, updating, revoking and looking-up digital identities. However, traditional centralized identity management systems on the Internet, relying on the so-called trusted third parties, raise many privacy concerns. For instance, most of the Identity Management (IdM) solutions are under the assumption that users and service providers trust the IdP, which increases threats from internal attacks of the IdM and hence compromises users privacy. Secondly, the traditional IdM systems suffer some longstanding security vulnerabilities and attacks such as single-point failure or phishing [5]. Facts (e.g., Equifax data breach [6], Facebook security breach [7], etc.) prove that centralized identity management systems have become honey pots of attackers. Thirdly, the proliferation of online identity providers also leads to fragmented identities scattered all over the Internet, which makes us overwhelmed by multiple accounts. Furthermore, fragmented identities from different security domains extremely increase the cost of identity identification and expose personal information retained by identity providers to vulnerabilities and data breaches.

With the advent of IoT, existing identity management systems on the Internet cannot be directly transplanted to IoT environments due to some native IoT characteristics such as scalability, interoperability, and mobility. These requirements are of great importance with an impact on the design of identity management systems for the IoT:**Scalability**: Internet of Things will comprise billions of individuals, collectives and everything in the cyber physical world, which demands highly scalable identity management. Apparently, using the traditional centralized IdM scheme where all IoT identities are maintained by one universal third party to build the highly scalable IdM solution becomes extremely unrealistic. Inevitably, there will be many different identity providers from different IdM systems. Albeit, the federated identity management solutions such as SAML [8] and Shibboleth [9], where different identities from different IdM systems can be managed, break the barriers between different IdM systems following the federation standards and successfully bring a silver lining of designing the IoT IdMS. However, in trustless networks, trust should be taken into account, that is how to build mutual trust relationships between different IdM systems. Consequently, identity management should be scalable and trusted in distributed trustless networks without a centralized control of any security authorities (i.e., identity providers, central access servers, etc.).**Interoperability**: The spectrum of objects makes them extremely heterogeneous with different communication, information and processing capabilities. Each object would be subjected to various technologies such wireless communication technologies (IEEE 802.15.4, WiFi, Bluetooth Low Energy, etc.), communication protocols (CoAP, LORA, MQTT, etc.), cellular communication technologies (GSM, UMTS, LTE, etc.) and hardware-dependent controllers (Arduino, Intel Edison, Raspberry, Eaglebone, etc.) [10]. Diversity and heterogeneity lead to interoperability problems, therefore, unifying all identities of IoT objects from different manufactures, vendors, communities and standard groups has been considered to be an impossible mission. Despite many recent initiatives, the emergence of standards remains highly fragmented, leading to divergence in vocabularies, methods and models (OneM2M [11], IoT reference architecture [12], etc.). The design of interoperable identity management of IoT objects remains balkanized without an integrative approach to make real progress in reducing software, hardware and communication heterogeneity.**Mobility**: The IoT is ubiquitous, which means IoT devices, such as vehicles or wearable devices, are subject to strong mobile capability. The mobility ensures users to connect services continuously even when moving [13]. No matter where these devices are located, we need to authenticate ourselves, get authorization and access controls to the corresponding device services [14]. Therefore, the identity management for the IoT should be characterized by mobility and provide a peer-to-peer authentication and authorization services.

Therefore, the Internet of Things calls for a brand new identity management paradigm to solve the existing identity security and privacy concerns on the Internet and take into account the native IoT unconventional characteristics.

In the following sections, we discuss the logic identity law from the philosophy origin, survey the art of digital identities on the Internet and analyze the traditional identity management solutions in the context of IoT. Then, we investigate the recent promising paradigm of the blockchain sovereign identity solutions, enumerate some projects and startups that are focusing on IoT identity problems, and finally point out challenges in building identity management systems for the Internet of Things.

## 2. The Law of Identity in the Philosophy of Logic

Identity was firstly formalized by Aristotle’s Law of Identity in logic as: “each thing is identical with itself”. Coupled with the Law of Contradiction and Law of Excluded Middle, the conclusion that identity is an equivalence relation with the characteristics of reflexive, symmetric and transitive, could be drawn [15]. Later, Wilhelm Gottfried Leibniz formulated the Leibniz’s Law, namely Identity of Indiscernibles [16] as: ”No two objects have exactly the same properties”. Consequently, the two following principles derived from the Leibniz’s Law, the Indiscernibility of Identicals) (Principle 1) and the Identity of Indiscernibles (Principle 2) are used for distinguishing two different individuals in the physical world and the cyberspace of the Internet due to the intuitive and simple recognition.

**Principle** **1.**
*For any x and y, if x is identical to y, then x and y have all the same properties.*
(1)∀x∀y[x=y→∀P(Px↔Py)]


**Principle** **2.**
*For any x and y, if x and y have all the same properties, then x is identical to y.*
(2)∀x∀y[∀P(Px↔Py)→x=y]


However, there are many paradoxes of the previous formalized definition. The most well-known paradox is the Ship of Theseus [15], where people cannot tell whether the two wooden ships are the same one with replacing and reassembling of the planks and beams. In detail, suppose all the planks and beams have been replaced through continuous repairing work over time. From the perspective of Principle 1, if the new ship (Ship B) is identical to the old ship (Ship A), they should have the same properties (planks and beams). However, the truth is they do not have the same planks and beams. Another unexpected situation is: if the replaced planks and beams are reassembled into Ship C, we can barely say, the previous Ship A and current Ship C are the same one even if they are composed by the same planks and beams according to Principle 2. In other words, only relying on attributes is not able to identify an entity in IoT era where all individuals, organizations and things are interconnected. Unfortunately, many identity management solutions in the cyberspace, are based on the Leibniz’s Law, where individuals are identified using a set of attributes and authenticated using credentials such as passwords. Due to this definition, systems, especially financial service providers, need to follow the Know-Your-Customer (KYC) [17] procedure to verify the identity and store identities of its clients, which gives rise to fragmented identities and renders personal identity information theft even more severe. Because, in the Leibniz’s Law (attribute) based identity model, users cannot prove their identities unless providing sufficient sensitive personal information and every system becomes a stand-alone identity store (also known as an Identity Silo). Personal identity information is over-harvested by many untrusted service providers and hence easily stolen due to inadequate defense measures of some unreliable systems.

## 3. Traditional IdMS on the Internet

Digital identity management systems are responsible for managing users’ identity information, consisting of identifiers (UserID, email, URL, etc.), credentials (certificates, tokens, biometrics, etc.) and attributes (roles, positions, privileges, etc.) [4]. Figure 1 depicts a sample instance of traditional IdM systems, which comprises three main stakeholders: subject (also known as user), relying party (also called service provider) and identity provider (IdP) [18]. The three different parties are interdependent entities: the subject requests access to services from the relying party, which requires the identity provider to challenge the subject identity through the authentication protocol.

Since the birth of Information Technology, IdMS have always been regarded as the keystone to access services and resources on the Internet. In the course of the last three decades, IdMS have evolved from isolated to centralized and then to federated models [19].

In the Isolated IdM model, identity providers have played a central roles as relying parties (service providers) by providing subjects (users) access to Internet services and resources maintained by a single security domain [19]. When subjects decide to access Internet services, the first step is to register themselves to service providers and obtain digital identities with credentials from their security domain. Nevertheless, the rapidly proliferating of online services in various security domains incur the identity bloating. It becomes an impossible mission for humans to manage many digital identities (e.g., memorizing their corresponding passwords) by following the isolated IdM model. To deal with this problem, the centralized IdM model aims at detaching identity management from service provision, and allows several service providers to rely on the same identity provider [20]. While the centralized IdM model reduces the number of user identities, users still require access to distributed services managed by different centralized IdM systems and security domains. The federated IdM model attempts to establish trust relationships between identity providers by which it becomes possible for users in one security domain to access services from another domain [21]. For example, Shibboleth [9], a federated identity management system for the Web, allows users to sign in to a security domain using just one identity and grant access to various systems belonging to the same federation of different organizations or institutions. The federated identity allows the sharing of information about users from one security domain to the other domains in the federation. Which means that no matter which identity is authenticated in one domain, services provided by another domain in the same federation are accessible based on credentials provided by its domain. However, the access of many unauthenticated third-party service providers to the detached identity providers could also cause the spread of phishing attacks.

The emergence of centralized and federated IdMS indeed alleviates the complexity of managing many identities originated from different security domains. However, the increasing number of applications per domain renders all agreements, protocols, standards, and processes (i.e., authentication and authorization) across these domains extremely complicated and undermines usability of identity. Besides, centralized and federated IdM systems are designed from the perspective of service providers [5]; they are still not flexible for lacking of users’ consideration. Therefore, user-centric identity management models have been proposed to improve user experiences and ensure security and privacy [22]. Jøsang and Pope [19], for example, proposed a user-centric IdM system where users manage their identities from different domains via personal trusted devices like smart cards or phones, which means that all user identities are maintained by few personal devices. OpenID [23] is also a user-centric and decentralized identity system for web services. It introduces ID token (JSON Web Token) on the basis of the OAuth 2.0 [24] authorization protocol to authenticate users. The decentralized framework makes the identity providers more robust for resisting DDoS (Distributed Denial of Service) attacks. However, the identity providers taking OpenID standard could see all the related web login information which also makes cross site tracking easier. Besides, the URL-based identifiers in OpenID usually compromises users’ privacy. Suriadi et al. [25] proposed an identity management system where they integrated federated Single Sign On (SSO) and followed user centric design principles to build the identity management, taking into account user privacy. In [26], Ahn et al. enhanced the privacy functionality in user-centric identity systems via applying privacy labels to personal claims so that the privacy of users could be protected according to different secrecy level. Although the user-centric identity management systems provide an improved solutions to manage identities of subjects and service providers, the trust assumption that users still have to put all the trust on the third-party identity providers is still in there and has not been eradicated. Users still have to rely on “trusted third-party” identity providers to access services in different domains while these identity providers could see all the transactions between users and service providers.

From the industrial perspective, we identify several initiatives and groups working on identity management in Table 1, including PRIMELife [27], SWIFT [28], DAIDALOS [29], Kantara [30] (Liberty [31]), FIDIS [32], SAML [8], Higgins [33], OpenID [23], Shibboleth [9], STORK [34], PICOS [35] and Cardspace [36] just to mention a few. We compare them according to the previous IoT identity management requirements including scalability, interoperability, mobility, security and privacy.

In Table 1, we can see that most of these systems cannot handle expansion: users have to consider all entities from IoT and coordinate different application domains to join the IdMS, which compromises scalability and increases the difficult to build an interoperable system in such heterogeneous environment. Although some initiatives such as OpenID or PICOS are capable of extensibility owing to the decentralized architecture to some extent, the IoT IdMS still needs a very strong extensible system for managing all entities in IoT environment. Besides, the mobile IdMS is critical as well under the environment of ubiquitous IoT devices or services. No matter where you are and where you move, the mobile IdMS should ensure the accessibility of user’s identities. Albeit, most of initiatives have the consideration in the security and privacy aspect. However, as mentioned before, they are also built on the common assumption that all users should trust their IdPs since the corresponding IdPs are definitely involved in every transaction, which undermines privacy of users. Similarly, despite adopting the user-centric model, these identity management systems still have to rely on third-party identity providers, which does not eliminate the privacy concerns.

To sum up, current identity management systems on the Internet have evolved from the Leibniz’s Law (attribute) based identity management model used in isolated systems to the decentralized, federated and user-centric identity management solutions such as OpenID adopted by many online service providers such as Facebook and Google. Usually, the IdMS of some big online service providers becomes the universal identity provider in their federated domains. In fact, the federated identity solution establishes some certain relationship among online service providers. Therefore, we could login in other online services only using one account such as our Facebook or Google account. Admittedly, the federated user-centric IdMS using relationships indeed alleviates the complexity of users managing their identities. However, security and privacy have not been solved perfectly in that users have to put all the trust on their identity providers who are sitting in the middle and can see all the activities between every user and their online service providers. Therefore, eliminating the unnecessary third parties and building a trusted identity provider in trustless networks are pretty critical for the IoT environment.

## 4. The Rise of Blockchain Identity Solutions

Over the past decade, blockchain technology, originated from Bitcoin [37], has raised a lot of attention due to the ability of eliminating intermediaries in transactions. The blockchain trend of thought, especially from Ethereum [38] in 2014, has spread to many realms from monetary and finance to governance, copyright and even IoT. As cyber security threats are becoming major challenges in IoT [39], blockchain technology is emerging as a prominent perspective to develop IoT security solutions in decentralized and trustless environments. In fact, blockchains can remove the intermediaries, and allow users and devices to manage their own identities without relying on third parties.

### 4.1. Blockchain Technology

Blockchain technology, which was initially introduced by the Bitcoin crypto-currency [37], opens opportunities to multiple initiatives and research topics in the context of IoT security [40,41]. Moreover, blockchains, as distributed ledgers, keep permanent records of all transactions used to transfer bitcoin values between members, participating in the Bitcoin peer-to-peer network. They also define the structure of how transactions should be organized into blocks, mined, confirmed and stored. Mining is the mechanism that allows the blockchain to be decentralized and secure. Nakamoto introduced the concept of Proof of Work (PoW) as a mining process to ensure consistency of transactions and solve the double spending problem in decentralized networks [37]. With the PoW blockchain, however, there is no need for any kind of a trusted authority, such as a bank, to keep track of the money transfer, all members have their own tamper-proof copy of the blockchain ledger. Each node in the Bitcoin peer-to-peer network maintains a copy of the blockchain. In addition, the blockchain is simultaneously updated through the peer-to-peer network so all members can validate any transaction instantly.

Since 2014, blockchain entered the 2.0 era led by Ethereum [38], which is a decentralized platform based on blockchain technology. It aims at creating a general purpose decentralized computer via the Turing-completeness smart contract concept, which allows writing and deploying all kinds of decentralized applications (Dapps) without any possibility of downtime, censorship or fraud. Buterin [38] explained Ethereum as: “combining the cryptographic algorithms with the economic incentives to create a decentralized network with memory.” To sum up, the blockchain provides us a new perspective to reconstruct the Internet in distributed P2P networks without any unnecessary intermediaries. It is the concept of eliminating central authorities and intermediaries through blockchain technology that opens opportunities to multiple initiatives and research topics in the context of IoT security [42]. Therefore, we provide a survey of how blockchain could transform the identity management systems.

### 4.2. Elucidation of Identity and Naming Systems

The distinction between identity management systems and naming systems (i.e., Domain Name Service (DNS), active directory or URLs) is blurry in the context of the Internet. At first sight, there is a slight difference between these types of systems. Identity management systems are coupled with service providers and used to identify resources or users in a particular domain, whereas the naming systems are designed to identify computers across networks, resources and user accounts in companies or social networks. DNS records, URLs and user accounts are somehow identities at the same time.

Blockchains are already used to build identifier or naming systems. The Namecoin [43], for example, is a fork of the Bitcoin blockchain that provides domain naming systems functionality via binding the human-readable name and IP address. It is the first solution for the naming trilemma of the Zooko’s Triangle [44] on building a secure, decentralized and human-meaningful naming system. By modifying Namecoin, Certcoin [45] builds decentralized authentication system (PKI), which defines a set of key operations like registering, updating, verifying and revoking. Based on Certcoin, Authcoin [46] proposes a new alternative protocol for authentication using a flexible challenge–response schema to PGP in the context of the Web of Things. Its successor, the Blockstack [47] attempts to redesign the naming system and PKI authentication features using state machines. It also added the storage aspect to its blockchain-based system to construct a kind of new type of Internet resource identification, preserving privacy and including property rights. Fromknecht et al. [48] tuned the certcoin parameters to ensure the retention of identities where users could not register the same already-registered identity again.

Even though naming systems could be exploited as identity providers for individuals in specific domains, the goals of identity systems and naming systems are completely different. The former attempts to find a way to uniquely define individuals in the cyberspace, whereas the latter is responsible for routing via assigning a unique identifier to retrieve a user or object in the service domain. Rather than working on naming systems, many research teams and recent projects as introduced in the following part, are working on the digital identity problem based on blockchains. These blockchain based identity solutions are promising of becoming the critical component of future digital world infrastructures.

### 4.3. Blockchain-Based IdMS

From an academic research perspective, blockchain-based IdM systems are gaining lot of attention to propose new solutions for digital identities: Bassam [49] introduced a blockchain-based PKI and implemented the solution based on Ethereum smart contracts. In his work, he defined several identity related operations such as adding attributes, signing attributes and revoking signatures. More importantly, he also calculated the cost of different operations in Ethereum platform. Liu et al. [50] developed an identity management system based on Ethereum smart contracts through binding public key and user’s entity information. Besides the identity management part, they also redefined the token to fit their proposed reputation model to reflect the reputation of users. Axon [51] analyzed privacy requirements when designing decentralized PKI systems and proposed a blockchain-based PKI with privacy awareness. In addition to a set of operations such as registering, revocation and recovery, they introduced the concept of neighbor group to enhance the performance of privacy preserving. Augot et al. [52,53] modified the Bitcoin stack to build an identity management solution and introduce a zero knowledge proof called Brands selective disclosure scheme [54] to ensure anonymity of the identity at the same time. Hardjono [55] introduced a blockchain-based privacy preserving identity solution called ChainAnchor using zero-knowledge proof in a permissioned blockchain environment. In ChainAnchor, verified nodes have the privileges to write or process transactions and others could only read and verify transactions. All verified nodes are built on the tamper-resistant hardwares and form the privacy preserving layer to provide privacy protection services to users. Halpin [56] designed NEXTLEAP, a decentralized identity framework with privacy preserving features using blind signatures. Moreover, they used authentication services provided by their identity solution to build a more secure messaging application. Azouvi et al. [57] also proposed a privacy preserving identity solution using blind signatures. They set up a threat model, performed a security analysis and implemented their solution in Ethereum. In Table 2, we chronologically list these blockchain based identity management solutions, most of which are from academia, and give main features of them.

In addition, several startups and IT players are focusing on the development of identity systems, such as Uport [58], Shocard [59], Bitnation [60], Civic [61], Jolocom [62], Sovrin [63], Evernym [64], ID2020 [65], Ethereum Identity Standard ERC725/735 [66], and W3C Decentralized Identifiers (DIDs) [67] just to mention a few. Generally speaking, these blockchain identity solutions fall into one of two categories: identity solutions relying on permissionless public blockchain platforms (e.g., Uport and ERC725/735) and identity solutions having authenticated block producers of the permissioned identity blockchain (e.g., Sovrin). For example, Uport, which is a core component of the Consensys Ethereum ecosystem [68], aims at building decentralized applications to solve the digital identity problem. It mainly uses the Ethereum smart contract to design digital identity model, and ensures reliability and usability of identities through a set of operations (i.e., keys revocation and identities recovery). Sovrin takes a different approach and provides a complete full stack to manage identities from the distributed ledger to devices. It adds the identity layer for every entity on the Internet and operates as a global public utility designed to provide permanent, private and trustworthy identities. Sovrin establishes a public permissioned blockchain in a peer-to-peer network in which nodes are divided into authenticated validator nodes and observer nodes to ensure high performance and scalability. More importantly, the sovrin token is introduced to their framework as their incentives to power their transactions.

In general, the blockchain-based identity is also called the self-sovereign identity, which denotes an approach that transfers access control rights and management of identities from traditional identity providers to the edge under the control of identity owners. In other words, only owners have the right to dispose their identities, which blocks attacks from malicious third-party identity providers. Although there are two different methodologies, permissioned (e.g., Sovrin) and permissionless (e.g., Uport), to implement the blockchain self-sovereign identity solutions, the basic concepts could be summarized as:Identities of individuals (i.e., human beings) and collectives (e.g., companies, banks, governments, etc.) can be selectively stored in the blockchain without compromising privacy.Individuals and collectives issue claims to each other using these blockchain identities. Basically, claims are the endorsement by other individuals or collectives, which could be governments, banks, universities or even friends.

As shown in Figure 2, there are two individuals (Alice and Bob) and several collective examples (i.e., companies, banks, governments, and schools). The individual or collective identity, composed of the identity attributes and identity claims, could be gradually completed through the following steps:The individual or collective, for instance Alice, could generate and add as many identity attributes (identifier, public and private key pairs, biometrics, etc.) as she wants.The individual or collective will create the blockchain identity through submitting identity related information such as public keys and the corresponding signatures.The individual or collective could use the public and private keypairs, which are correlated to the mined blockchain identity, to issue claims.

In Figure 2, Alice could self-generate identity attributes and also receive the claims from her employer, bank, government, school and even her friend Bob. In different scenarios, Alice could give the necessary identity claims to identify herself or to demonstrate that she has some qualifications. For example, when applying for a job, Alice can give the her ID Card from the government and diploma from her university. After entering the company, she has to give her bank account to receive the salary.

### 4.4. A Sliver of Light

The self-sovereign blockchain based identity management systems eliminate unnecessary centralized identity providers through creating the blockchain identity on the blockchain platform, in which all users and service providers follow the identity consensus and hence could verify identities instead of blindly trusting in some big third-party identity providers. Moreover, the concept of claims in blockchain IdMS essentially is an extension of relationships in the federated identity management model. Claims, as the endorsement relation from others, are indispensable in the trustless distributed blockchain based IdMS, since individuals still do not trust (or know) each other, even if they could verify the real identities with privacy concerns. Roughly speaking, digital identities facilitate communication among entities (e.g., human) and access to applications and services. As a result, where there is no interaction (relation), there is no identity. From the human evolution perspective, the more complex social relations become, the more difficulties humans could tackle. The social attributes, which are inherently existing in human society, are indispensable for human beings. This law applies equally the cyberspace, which is composed of IoT entities. A social network is composed of social entities (such as individuals or organizations) and social relationships between these social entities. Consequently, the concept of social networks (namely, the combination of identity and relationship) could be introduced to solve trust issue in distributed identity management.

Recently, researchers from the academic community have started to apply the social network principles to the Internet of Things forming the Social Internet of Things (SIoT) [69]. The SIoT paradigm seeks a transition from isolated devices to friendly unified devices which could find friend devices and manage their relationships. By such, devices can be involved in social-like networks, in which devices can publish their services to improve visibility and find services. The relationships hence make devices smarter and able to interact without human interventions. However, the SIoT paradigm [69] only focuses on the social networks among devices and does not cover IoT security issues. In [70], the authors built a security architecture for IoT using blockchain based social networks, which not only unifies all IoT entities from people, organization, physical and virtual objects but takes into account IoT security problems using the social network of device owners. Specifically, they firstly classified IoT entities into two categories: subjects and things. The former stands for individuals (usually, human beings) or collectives (e.g., companies, governments, and organizations), who possess root identities that could be packed into identity transactions and sealed into blocks. The latter refers to IoT services provided by software applications or interfaces residing in IoT devices, which are designated by partial identities. Therefore, identities are divided into two types: root identities for subjects and partial identities for things. Then, the relationships of Subjects-to-Subjects (S2S) and Things-to-Things (T2T) and the ownership of Subject-to-Things (S2T) compose the three-dimensional social networks, in which root identities are managed by blockchains in trustless peer-to-peer networks and owners (root identities) could configure their access control security policies through the ownership between subjects and things. To sum up, the socialized IoT paradigm not only provides a solid foundation to resolve security and trust issues in the distributed trustless IoT environment but points out a new direction to solve the previous IoT challenges such as scalability, heterogeneity and mobility due to the inherent characteristics of social networks such as extensibility, human-centric, ubiquity, etc. [69].

## 5. Challenges in IdMS for IoT

Even though many promising solutions (i.e., blockchain-based identity management and socialized IoT paradigm) are proposed, other critical supporting components of building effective IdMS for IoT remain challenges such as [18] access controls, privacy, trust and performance.

### 5.1. Access Controls

Access control refers to a security mechanism which regulates who can access what kind of resources or services in computer systems. Identity and access control are always tied together. The goal of establishing identity systems in the IoT is to enable communications and properly regulate the process of authorization to devices and resources. In other words, identities point out a set of access permissions during the interaction between two subjects while, in computer systems, identities are used for access controls. Therefore, designing the access control mechanism should fully take into consideration the existing identity management model.

With the advent of IoT era, many traditional access control models such as Access Control Lists (ACLs) and Role-based Access Control (RAC) models [71], which are designed for centralized systems, become obsolete due to the rapid growth of roles and policies. Besides, more and more factors and parameters such as time and location should also be taken into consideration in designing access control solutions. Although Attribute-based Access Control (ABAC) model aims at handling this problem, the existence of centralized identity providers in ABAC model still have to face up to the scalability issue. A common problem of existing solutions stems from centralized administrative parties (i.e., administrators or identity providers) that become indispensable for assigning access rights, roles and attributes, and, consequently, these solutions are not suitable for scalable decentralized IoT systems. The Capability-based Access Control (CAC) model [72,73,74] raises a lot of attention by virtue of its flexibility. However, it has the same premise by which users who request services need to rely on the authentication of third parties such as identity providers or certificate authorities. This is apparently unsuitable in the trustless IoT environments where users could be generated by each subject (e.g., human) without the endorsement of other intermediate parties. In other words, the CAC model only works in trusted environments. From the discussion above, the major challenges of designing the access control mechanism for IoT could be summarized as:How can an effective access control mechanism be designed that could universally manage access permissions to various IoT entities (people, things, services, etc.) without worrying about the rapid growth of users, roles and policies?How can the access control mechanism in trustless IoT environment be built if the “trusted” third-party identity providers are removed using blockchain technology?

Therefore, none of the proposed access control models (ACLs, RAC, ABAC or CAC) can satisfy the scalable, interoperable and trustless IoT environment. The access control mechanism for IoT should be rebuilt based on these IoT characteristics, security and privacy premise. Recently, some researchers try to use blockchain technology to design access control systems for the Internet of Things [40,75,76]. They rely on blockchain transactions to grant or deny access requests in IoT environments through defining the access control transactions using smart contracts, which takes advantage of the immutability of the blockchain and makes the auditing of access control policies transparent. However, the blockchain is not a panacea for all kinds of problem. Since the storage resources are pretty scarce and expensive in public blockchain platforms, it is hard to imagine that access control policies of all IoT devices are uploaded to these blockchains. Although the private blockchain access control mechanism could solve the storage consumption problem, the isolated private blockchain systems will undoubtedly hinder the large-scale adoption due to the interoperability issue. Reference [70] proposed an architecture called FOCUS, in which they took advantage of the three-dimensional social networks: the social relationships between owners, social relationships between things, and the ownerships between owners and things to build the user-centric access control mechanism to universally manage all types of access control policies. Furthermore, the entire access control mechanism is built on a blockchain based identity management system in trustless IoT environment, which guarantees security and privacy of users. Specifically, in FOCUS, the blockchain is used for recording identities of owners and stores them on peers in trustless peer-to-peer networks. Then, through these identifiable owners’ blockchain identities, owners could assign access policies to things in the same ownership domain and the relationships between things could form a feedback loop to affect the formulation of access policies.

### 5.2. Privacy

The privacy preserving refers to the protection of users’ sensitive information such as identity information, location, mobility traces, and habits from any other parties. From users perspective, the privacy preserving includes two aspects:Identity information protection from identity providersSensitive application data protection from service providers

Many works [77,78,79] in academic papers and IT industry are proposed to preserve the sensitive application data rather than identity information stored in identity providers. In most cases, identity providers and service providers are bounded together and they require some personal information to authenticate users. For instance, users may protect their location information from map service providers by disabling the location service, however, they ignore the leakage of their personal identity information by identity providers exposed to security vulnerabilities. Although these proposed solutions [77,78,79] solve the privacy problem to some extent, their identity information is still exposed to identity providers.

Before blockchains, privacy preserving is incomplete owing to the existence of centralized identity providers in that the identity information protection remains unsolved. Service accessors and service providers need to grant full trust to their identity providers. In other words, centralized identity providers could see activities between service accessors and service providers, which compromises the identity information privacy. Fortunately, the self-sovereign blockchain identity management is taking the control right of identities back to users from the third-party identity providers. The design of identity solutions are thus subjects to a paradigm shift by which users decide to whom their sensitive personal information could be revealed (from user’s perspective) instead of trusting identity providers to manage their personal information. Although users could have full control over their personal information in blockchain based identity management systems, the public blockchains can still expose some identity information. Therefore, we still need to introduce the privacy preserving scheme in that the introduction of privacy preserving tools such as the multi-party computation [80] or zero-knowledge proofs [55,81,82] into blockchains could bring the selective disclosure of sensitive personal information and perfect online identity privacy into reality. Privacy preserving has always been one of the hottest topics in blockchain industry. Since 2009, academic researchers have proposed many privacy preserving schemes including Pinocchio [83], zk-Garbled Circuits [84], zk-SNARKs [85], ZKBoo [86], zk-STARKs [87], and Bulletproofs [88]. The goal of these schemes is to design a succinct non-interactive zero knowledge protocol to protect the privacy in blockchain systems. However, due to the complexity of these schemes, only zk-SNARKs and Bulletproofs are deployed in Zcash [89] and Monero [90] systems, respectively. As for the integration with blockchain smart contracts, it still needs many effort from academia and industry.

### 5.3. Trust

The trust management is closely related to the identity and access control management framework. Although there is no consistent definition on trust [91], researchers recognize the importance of the trust management. Many schemes have been proposed to manage trust in the context of IoT in order to deal with misbehaving IoT devices. Reference [92] presented a fuzzy reputation based trust management solution for IoT wireless sensor networks, in which they considered the packet forwarding/delivery ratio and energy consumption as the Quality of Services (QoS) metrics to evaluate the trust relation. Recently, the SIoT [69] obtains much attention due to the high extensibility through integrating social networks concept to IoT. Therefore, many trust management solutions have emerged based on SIoT paradigm. For instance, authors in [93,94,95] introduced trust management solutions for SIoT paradigm. Reference [96] presented a similar trust management protocol for IoT with no centralized trusted authority. All these solutions make use of not only the QoS metrics but also the ownership between devices and owners and social relationship between users to evaluate the trust to defend against attacks.

However, these trust management solutions are also built on the previous implicit identity assumption that users and service providers should put all trust to their identity providers so that they could identify each other in the same security domain. Within the same security domain, users and service providers trust and rely on the same identity provider, admitting that their personal information will not be compromised or exploited by the identity provider or third parties. In many cases, identity providers are subject to vulnerabilities which expose personal information repositories to be stolen by deliberated attackers (e.g., Equifax data breach [6] and Facebook security breach [7]). However, the implicit trust in identity providers becomes questionable with the increasing attacks [6,7]. Undoubtedly, blockchain based identity management systems eliminate the unnecessary information exposure to third parties and provide many good characteristics such as immutability, neutrality and secure timestamping which could be used for building trust relationships. For instance, Lu et al. [50] presented an interesting approach of building the trust reputation via tailored Ethereum tokens. Zhu [3] combined the blockchain and the social networks between all IoT entities to build a security architecture for IoT, which apparently lays a solid foundation for the trust management. However, these decentralized or distributed ways still face up to many difficulties in building the reputation system or feedback mechanism for aggregating trust relations on each parties including all subjects and service providers. The applications should be redesigned in decentralized way and be autonomously decentralized applications (Dapp).

### 5.4. Performance

Performance evaluation has been the essential scale of identity management systems. IdMS evaluation relies on the performance of servers running the identity management system, which will not work in distributed IoT environment composed of billions or even trillions objects. Therefore, building the metrics for distributed identity management systems becomes important for performance evaluation. Frameworks of evaluating the distributed systems such as in [97] are indispensable in order to measure the performance of distributed systems. However, following the theory in [97] that the scalability of distributed systems can be measured by the ratio of two productivity-based functions *F*, we still need to refine and designate some parameters to the predefined general family of metrics, which include the rate of providing services (λ), the quality of services (QoS) and the cost of providing services (*C*) in the following equation:(3)$$\psi =\frac{F\left(\lambda_{2},QoS_{2},C_{2}\right)}{F\left(\lambda_{1},QoS,C_{1}\right)}$$
to build the evaluating framework for distributed identity management systems.

Similarly, the blockchain gives us the ability to design a distributed identity system in trustless environment, however, the performance of blockchain based systems still needs to be measured and quantified. Evaluating the performance of a blockchain system, in essence, refers to evaluating scalability of the distributed systems, which needs a holistic consideration about the cost and Quality of Service (QoS). As blockchain continuously gains interest in academia, some researchers propose their analysis framework for blockchain systems. For example, Gervais et al. [98] analyzed proof-of-work based blockchain systems such as Bitcoin, Litecoin, and Ethereum with different operational parameters. Dinh et al. [99] proposed their evaluation framework for private blockchain systems, in which they analyzed blockchain systems from the consensus, data model, execution layer and application layer. They also quantified the system from specific metrics including throughput, latency, scalability and fault tolerance. However, these proposed evaluation metrics for blockchain systems still need to be adjusted due to the particularities of blockchain based identity management systems. For instance, most of the transactions are identity lookup operations, which could be designed for free offline transactions without fees [3] in permissionless blockchain platforms such as Ethereum or Bitcoin. Onchain identity operations such as registration, revocation and update are quite rare compared to the offchain identity lookup operations. Therefore, the scalability evaluation of blockchain based IdMS is not identical with the previous mentioned evaluation methodology [98,99]. In general, performance evaluation framework for distributed identity management systems especially under the blockchain circumstance needs to be developed.

## 6. Conclusions

In this survey, we explore the origin of identity and sort out a comprehensive development roadmap of identity management systems on the Internet. Combining native IoT characteristics such as scalability, interoperability, mobility, security and privacy, we point out the deficiencies of the traditional IdMS on the Internet. Although there are still some challenges in building a feasible and effective IdMS, the emerging of blockchain technology indeed sheds light on building an independent, neutral and secure identity management solutions in IoT. 

## Figures and Tables

**Figure 1 sensors-18-04215-f001:**
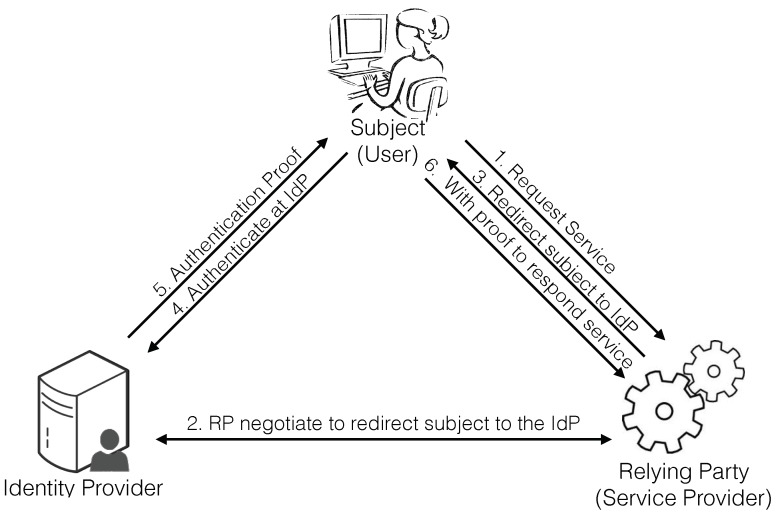
Stakeholders from the traditional IdMS model.

**Figure 2 sensors-18-04215-f002:**
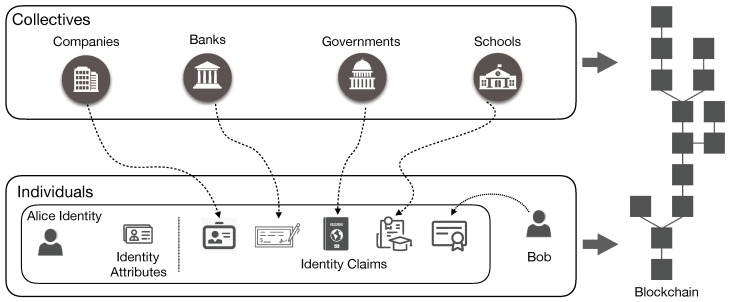
Overview of blockchain based identity management solutions.

**Table 1 sensors-18-04215-t001:** Identity management initiatives comparison.

	Scalability	Interoperability	Mobility	Security & Privacy	User-Centric
PRIMELife(PRIME)			*	*	*
SWIFT(DAIDALOS)		*	*	*	*
Kantara(Liberty)	*	*	*	*	
FIDIS		*	*	*	*
SAML		*			
Higgins		*		*	*
OpenID	*				*
Shibboleth				*	
STORK		*		*	*
PICOS	*	*	*	*	*
Cardspace		*	*	*	*

**Table 2 sensors-18-04215-t002:** Blockchain identity management solutions.

	DNS	PKI	Storage	Bitcoin Based	Ethereum Based	Full Stack	Reputation	Privacy	Year
Namecoin	*			*					2014
Certcoin	*	*		*					2014
Fromknecht	*	*		*					2014
Uport		*			*				2015
Sovrin		*				*	*	*	2016
Jolocom		*			*				2016
Blockstack	*	*	*	*					2016
Authcoin	*	*		*					2016
ChainAnchor		*				*		*	2016
Liu et al		*			*		*		2017
NEXTLEAP		*						*	2017
Azouvi		*			*			*	2017
Axon		*		*				*	2017
Augot		*		*				*	2017
SCPKI		*			*				2017

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
