# Peer review of "Identity Management Systems for the Internet of Things: A Survey Towards Blockchain Solutions"

_sensors, 2018, doi:10.3390/s18124215_

Reviewer 1 Report

Please explain why we need Identity Management (IdM) for IoT devices.

In page 2, line 44. The authors claim that IoT has three characteristics: scalability, interoperability, and mobility; and In page 3, line 103. Digital Identity Management systems are responsible for managing users’ identity information. The authors should define what’s the characteristics for a IoT device, and explain why an IoT device need digital identity management.

Author Response

We uploaded the attachment file as our response.

Reviewer 2 Report

The paper investigates a very interesting topic in the IoT scenario: the identity definition and management. In the abstract, the authors declare that no other surveys/work have been provided in literature on the requirements of IdMS. However, the paper presents important flaws, explained below.

In the introduction, the considerations about interoperability and mobility should be supported by proper references.

Section 2 presents contents which are not so significant with respect to practical analysis and development of IdMS. In the reviewer’s opinion it must be omitted.

The solutions presented in Section 4.2 and 4.3 must be further detailed; the reader should clearly understand how such solutions work effectively. Moreover, the introduction of SIoT in Section 4.4 is not well motivated.

Section 5.1 does not reveal challenges in IdMS. The same is for the next sections; in particular, in Section 5.2 it appears that blockchain solves all the privacy issues. The aim of the survey should be pointing out the issues/challenges/directions of the research in IdMS. Hence, the authors must provide a more insightful analysis about such topics.

Author Response

Response to Reviewer 2 Comments

Point 1:  In the introduction, the considerations about interoperability and mobility should be supported by proper references.

Response 1:

Interoperability paragraph Page 2 (3 references added)

·       Add the missing reference (line:70) for various communication technologies.

·       Add two missing references (line: 74) of OneM2M and IoT reference architecture

Mobility paragraph Page 2 (2 references added)

·       Add a sentence and cite the corresponding reference (line:79-80)

·       Add a missing reference (line:82)

Point 2: Section 2 presents contents which are not so significant with respect to practical analysis and development of IdMS. In the reviewer’s opinion it must be omitted.

Response 2: Section 2 explores the origin of identity problem, in which we point out

·       Most identity management solutions, especially in the cyberspace are based on the Leibniz's Law, where individuals are identified using a set of attributes and authenticated using credentials like passwords.

·       The definition of Leibniz's Identity Law gives rise to fragmented identities and renders the identity theft even more severe in current identity management systems

We noticed the formulation of this section was not very clear and not easy to understand. Therefore, we made the following

Modifications:

·       Change the tittle (adding from the Logic to Blockchain Solutions), readers could easily find the paradigm shift from the old Leibniz's Law (attribute) based identity solution to the blockchain solutions.

·       Add more details about the Leibniz's Law -- Ship of Theseus paradox (line:103-111)

·       Add the issues (fragmented, over-harvested identity) caused by Leibniz's (attribute) based identity solution (line:114-121)

·       Add a transition from Leibniz's Law based Identity solutions to the current identity management solutions using federated relationships at the end of section 3 (line:197-203)

·       Add the problems of current IdMS in the Internet: security and privacy at the end of section 3 (line:204-207)

Point 3: The solutions presented in Section 4.2 and 4.3 must be further detailed; the reader should clearly understand how such solutions work effectively. Moreover, the introduction of SIoT in Section 4.4 is not well motivated.

Response 3: In order to help readers to understand how the blockchain identity solutions work, at the end of section 4.3, we summarized the current blockchain based identity solutions and illustrated the basic concept of these blockchain based identity solutions from the new Fig.2

Modifications:

Section 4.3:

·       Add two more important identity initiatives (line:297)

·       Classify the blockchain identity solutions into two categories: permissioned and permissionless (line:298-301)

·       Explain the self-sovereign identity and how it works (line 312-339)

·       Add Fig.2

Section 4.4: we noticed the previous section 4.4 is not self-contained enough, therefore, we reorganized this section as the summary of blockchain based identity solutions.

·       Move the first paragraph to Section 4.3 (line:341-346)

·       Add the summary to blockchain IdMS:  (line: 352-353) (line: 369-383)

1.     emphasize the importance of eliminate the unnecessary third party identity providers

2.     analyse the effect of claims and explain why claims are so important

3.     give two reference examples to support the 2nd point

Point 4: Section 5.1 does not reveal challenges in IdMS. The same is for the next sections; in particular, in Section 5.2 it appears that blockchain solves all the privacy issues. The aim of the survey should be pointing out the issues/challenges/directions of the research in IdMS. Hence, the authors must provide a more insightful analysis about such topics.

Response 4: we realized the problem and made the following

Modifications:

Section 5.1

·       Add a transition between identity and access control through

1.     explaining definition of access control (line:393-394)

2.     pointing out “the identity and access control always tie in together” and previous content as the reason (line:394)

3.     drawing a conclusion: designing access control should take into account the existing identity management solutions (line: 398-399)

·       add the challenges when building access control for IoT (line:414-420)

·       summarize the traditional access control models and point out the future direction by adding more references. (line: 421-446)

Section 5.2: In this section, we define the two aspects of privacy preserving, point out the current works are mainly focusing on one aspect (application data privacy) and give the new direction and challenges of developing new privacy preserving solutions under the blockchain circumstance.

·       Add sentence to explicitly present the challenges of previous privacy preserving solutions. (line:462-467)

·       Rephrase some sentences to make the future direction (Blockchain) more clear. (line: 467-469)

·       Add advantages and blockchain privacy challenges (line: 471-474)

·       Introduce the privacy preserving tools to solve the privacy problem (line: 474-477)

·       Add the challenges to integrate privacy preserving tools to blockchain. (line: 478-484)

Section 5.3:

·       Add some related previous work references (line: 486-498)

·       Add sentences to explain the challenge (line: 499-501)

·       Delete irrelevant sentences (line: 507-514)

·       Add the reference to explain the future direction (line:518-520)

Section 5.4:

·       Rephrase sentences to make challenges clearer (line: 526-533)

1.     How to define evaluation metrics of distributed systems

2.     How to evaluate the blockchain based systems

·       Add transition words to the future direction of references (line: 548-556)

Reviewer 3 Report

- I feel that the ultimate purpose of the survey is to introduce and survey the blockchain-based IdM systems, therefore the title should be changed to an appropriate one.

- If possible, please add one more section about how blockchain is leveraged in IdMS in details, including some figures illustrating system architecture and procedures in the Blockchain-based IdMS. Based on that, authors can analyse pros and cons of the current solutions.

Author Response

(The authors gave the same response as above.)

Reviewer 4 Report

This is a very well written and interesting paper. I would recommend the paper to be accepted  after major revision.

Follow some comments (in no particular order):

Figure 1 should probably appear on page 3.

In 5.4 explicit the theory in [73].

Through the whole paper should "the blockchain system" be "a blockchain system". There is several implementations around after all.

It would be good to give an exemplary blockchain identification architecture (e.g. a figure + text) before diving into section 5, so readers less familiar with the topics of blockchain can clearly understand how it is used. At this point for an uninitiated reader, the paper looks a bit too abstract.

Similarly, I would expect section 5 to dive a deep more into technical details so I can understand clearly what is the problem they try to solve and how they solve it. At the moment this section is very high level and does not provide enough information.

Again, it would be good to details what blockchain-based access control is in practice and to contrast it with other existing approach more clearly.

To summarise, I believe this is very interesting work. This is an article worth publishing. However, I strongly encourage the authors to make the paper more self-contained. Most of the work discussed is presented at a too high-level of abstraction. After reading a survey, I expect to have gained a basic level of understanding of the surveyed topic. Here, I feel more that I have been given a list of paper I should read.

Author Response

We uploaded the attachment file as our response.

Round  2

Reviewer 2 Report

The overall quality of the paper has been improved in the new version of the paper. I have no further comments.